Phage on tap–a quick and efficient protocol for the preparation of bacteriophage laboratory stocks

Bonilla Natasha
Rojas Maria Isabel
Netto Flores Cruz Giuliano
Hung Shr-Hau
Rohwer Forest
Barr Jeremy J. jeremybarr85@gmail.com
Department of Biology, San Diego State University , San Diego, CA , United States
Wiles Siouxsie
Electronic publication date: 2016 Jul 26
Publication date: 2016
Volume: 4
Electronic Location ID: e2261
Received 2016 Apr 16; Accepted 2016 Jun 26
Copyright: © 2016 Bonilla et al.
Copyright year: 2016
Copyright holder: Bonilla et al.
License: This is an open access article distributed under the terms of the Creative Commons Attribution License, which permits unrestricted use, distribution, reproduction and adaptation in any medium and for any purpose provided that it is properly attributed. For attribution, the original author(s), title, publication source (PeerJ) and either DOI or URL of the article must be cited.
License URL: https://creativecommons.org/licenses/by/4.0/

Keywords: Bacteriophage, Endotoxin, Cesium chloride, Top agar, Ultrafiltration, Dialysis, Speed vacuum, Phage bank

Funding: National Institutes of Health R01GM095384-01 to F.R. National Institute of General Medical Sciences NS047101 and R21AI094534 Gordon and Betty Moore Foundation Investigator Award 3781 This work was supported by the National Institutes of Health (Grant R01GM095384-01 to F.R.), Grants NS047101 and R21AI094534 from the National Institute of General Medical Sciences, and the Gordon and Betty Moore Foundation (Investigator Award 3781). J.J.B. received funding and support from San Diego State University. The funders had no role in study design, data collection and analysis, decision to publish, or preparation of the manuscript.

==============================
A major limitation with traditional phage preparations is the variability in titer, salts, and bacterial contaminants between successive propagations. Here we introduce the Phage On Tap (PoT) protocol for the quick and efficient preparation of homogenous bacteriophage (phage) stocks. This method produces homogenous, laboratory-scale, high titer (up to 1010–11 PFU·ml−1), endotoxin reduced phage banks that can be used to eliminate the variability between phage propagations and improve the molecular characterizations of phage. The method consists of five major parts, including phage propagation, phage clean up by 0.22 μm filtering and chloroform treatment, phage concentration by ultrafiltration, endotoxin removal, and the preparation and storage of phage banks for continuous laboratory use. From a starting liquid lysate of > 100 mL, the PoT protocol generated a clean, homogenous, laboratory phage bank with a phage recovery efficiency of 85% within just two days. In contrast, the traditional method took upwards of five days to produce a high titer, but lower volume phage stock with a recovery efficiency of only 4%. Phage banks can be further purified for the removal of bacterial endotoxins, reducing endotoxin concentrations by over 3,000-fold while maintaining phage titer. The PoT protocol focused on T-like phages, but is broadly applicable to a variety of phages that can be propagated to sufficient titer, producing homogenous, high titer phage banks that are applicable for molecular and cellular assays.

Introduction

Due to increasing interest for the use of bacteriophage (phage) in medical, industrial, and molecular settings, new approaches are required to quickly and efficiently produce high titer, homogenous, and purified phage stocks. It is desirable that these stocks be free of bacteria, molds, debris, culture medium, and bacterial endotoxins (Adams, 1959). Typically the ability to produce high titer phage stocks is largely dependent on the particular phage and host cell under consideration, yet certain principles and methodologies can be broadly applied. Traditional techniques used for the concentration and purification of phage involve centrifugation, filtration, ultrafiltration, precipitation with Polyethylene Glycol (PEG), ultracentrifugation in cesium chloride (CsCl) gradients, and dialysis (Adams, 1959; Yamamoto et al., 1970; Seeley & Primrose, 1982; Suttle, Chan & Cottrell, 1991; Carlson, 2005; Bourdin et al., 2014). Yet many of these techniques are time consuming and affect phage recovery and/or viability.

Phage preparations are often contaminated by macromolecules derived from the host bacteria and culture media, with the major pyrogen being the lipid A moiety of lipopolysaccharide (endotoxin) from the outer membrane of Gram-negative bacteria (Raetz et al., 2007). Endotoxins are amphipathic molecules; the lipid component is linked to a core polysaccharide and as a result they can form large aggregates greater than 1,000 kDa in solution (Magalhães et al., 2007). Endotoxin elicits a wide variety of pathophysiological effects in the body. Exposure to even small amounts can result in toxic shock, cell injury, cytokine production and the activation of immune responses (Morrison & Ulevitch, 1978; Rietschel et al., 1994; Alexander & Rietschel, 2001; Pabst et al., 2008). Due to these effects it is important that endotoxins be removed from phage preparations when studying or applying phage in the context of eukaryotic systems. The amount of endotoxin is defined as an endotoxin unit (EU), which corresponds to the activity of 100 pg of E. coli lipopolysaccharide. The endotoxin content of distilled water is estimated at 20 EU·ml−1, with the allowed limit for intravenous and oral administration set at 5 EU·kg·hr−1 and < 20 EU·ml−1 respectively (Bruttin & Brüssow, 2005; Gorbet & Sefton, 2005; Abedon et al., 2011).

A further limitation with traditional phage preparations is the variability of titer, salts, and bacterial contaminants produced between successive propagations. Eliminating this variability has proven critical for accurate analysis for the molecular interactions of phage within the context of eukaryotic hosts (Barr et al., 2013; Barr et al., 2015). Here we present the Phage On Tap (PoT) protocol as a fast and efficient way to produce homogenous laboratory phage stocks. Phage stocks were sterilized by centrifugation, 0.22 μm filtration, and chloroform treatment, before concentration and washing using ultrafiltration, and storage at 4 °C for four months with minimal degradation. However, the PoT protocol was not able to effectively reduce bacterial endotoxins, likely due to our ultrafiltration-based approach and the large aggregate size of endotoxins. Numerous endotoxin removal procedures and commercial kits are available (Boratyński et al., 2004; Merabishvili et al., 2009; Oślizło et al., 2011; Magalhães et al., 2007; Branston, Wright & Keshavarz-Moore, 2015), yet many of these methodologies lack generality, are time consuming, or are cost-prohibitive. Recently, the successful reduction of endotoxins (< 20 EU·ml−1) from phage lysates was reported by extraction with organic solvents (Szermer-Olearnik & Boratyński, 2015). We corroborate this organic solvent-based method for the removal of bacterial endotoxin from phage lysates and adapt this methodology, with reduced processing time through the use of speed vacuum, in the PoT protocol. The PoT protocol can purify phage lysates, with volumes ranging between 50–300 ml (or greater if required), producing homogeneous phage stocks for further laboratory testing. The method takes two days to purify and concentrate a final phage lysate of ∼10 ml of with a titer > 1010 PFU·ml−1, and a significant reduction in bacterial endotoxin levels. Our method has been tested on a variety of phages, including T4 (E. coli), T3 (E. coli), T5 (E. coli), and Spp1 (B. subtilis), and is broadly applicable to other tailed phages.

Materials and Methods

Reagents

- Luria-Bertani (LB) broth (cat. no. DF0446; Fisher Scientific)

- Agar (cat. no. BP1423; Fisher Scientific)

- Calcium Chloride Dihydrate (CaCl2·2H2O) (cat. no. C69; Fisher Scientific)

- Magnesium Chloride Hexahydrate (MgCl2·6H2O) (cat. no. BP214; Fisher Scientific)

- Sodium Chloride (NaCl) (cat. no. S671; Fisher Scientific)

- Magnesium Sulfate Heptahydrate (MgSO4·7H2O) (cat. no. M63; Fisher Scientific)

- Trizma Hydrochloride (Tris HCl pH 7.4) (cat. no. 93313; Sigma-Aldrich)

- Chloroform (cat. no. BP1145; Fisher Scientific) CAUTION: Chloroform is toxic and should only be used in a fume hood and with personal safety equipment, such as gloves and goggles.

- Pierce™ LAL Chromogenic Endotoxin Quantitation Kit (cat. no. 88282; Thermo Fisher)

Note: Endotoxin Quantitation Kit is optional and is only required for the quantitation of endotoxins in phage lysates.

- Ethanol, Absolute (C2H5OH) (cat. no. BP2818; Fisher Scientific)

- Glycerol (cat. no. G31; Fisher Scientific)

Equipment

- 37 °C incubator with a rocker

- Centrifuge with swinging bucket rotor

- Microtube centrifuge

- Centrifugal vacuum concentrator

- Stir plate

- Petri dish with disposable lid (cat. no. 09-720; Fisher Scientific)

- 1.7 ml microcentrifuge tubes (cat. no. 02-681; Fisher Scientific)

- Falcon 50 ml conical centrifuge tubes (cat. no. 14-432; Fisher Scientific)

- Small glass test tubes 13 × 100 mm (cat. no. 14-958; Fisher Scientific)

- Serological pipettes (cat. no. 07-200; Fisher Scientific)

- 0.22 μm Sterivex filter units (cat. no. SVGP; Millipore)

- Whatman Anotop 0.02 μm sterile syringe filters. (cat. no. 09-926-13; Fisher Scientific)

- Amicon® Ultra-15 centrifugal filter units, Ultracel 100 kDa membrane (cat no. UFC910008; Millipore)

Note: 100 kDa membrane pore size is equivalent to a spherical particle with a diameter of ∼3 nm and is therefore sufficient for the capture of all known bacteriophages (Erickson, 2009).

- 500 ml PYREX® screw cap storage bottle with plastic seal (cat. no. 13-700-446; Fisher Scientific)

- Spectra-Por® Float-A-Lyzer® G2 Dialysis membrane, 10 mL, MWCO 3.5–5 kDa (cat. no. Z726273; Sigma-Aldrich)

- Nalgene™ General Long-Term Storage Cryo Tubes (cat. no. 03–337; Fisher Scientific)

Reagent setup

LB broth: 25 g LB broth in 1 liter dH2O.

LB top agar: 25 g LB broth, 7.5 g Agar, in 1 liter dH2O.

LB agar plates: 25 g LB broth, 15 g Agar, in 1 liter dH2O.

SM buffer: 5.8 g NaCl, 2.0 g MgSO4·7H2O, 50 ml 1 M Tris-HCl pH 7.4, in 1 liter dH2O. Autoclave, 0.02 μm filter-sterilize before use, and store at room temperature.

Calcium chloride (CaCl2): Prepare a 1 M stock solution and add a final concentration of 0.001 M to desired volume of the LB broth that will be used for the liquid lysate. Autoclave, 0.02 μm filter-sterilize before use, and store at room temperature.

Magnesium chloride (MgCl2): Prepare a 1 M stock solution and add a final concentration of 0.001 M to desired volume of the LB broth that will be used for the liquid lysate. Autoclave, 0.02 μm filter-sterilize before use, and store at room temperature.

Phage on Tap (PoT) Protocol

Notes: The procedure for phage propagation is largely specific for each phage and bacterial host. Here we use propagation conditions for T4 phage and Escherichia coli B bacterial host. It is recommended to use appropriated growth and propagation conditions for your choice of phage and host.

Once a sufficiently high titer phage lysate is obtained please proceed to step 3.

It is recommended to only propagate and purify one phage at a time to prevent cross-contamination.

1| Phage plaque assay for determination of titer (Adams, 1959)

Grow E. coli B bacterial host in LB broth overnight at 37 °C.

Dilute phage stock or isolate in LB broth down to the desired dilution (e.g., for a phage stock of 108 PFU·ml−1 serially dilute down to 10−6 and 10−7 to obtain countable plaques).

Heat LB top agar in microwave until completely molten, then allow top agar to cool in a 56 °C water bath or until it is warm to the touch.

Add 1 ml of the overnight bacterial host and 1 ml of the phage dilution to a glass test tube and mix.

Add 3 ml of molten top agar to the glass test tube.

Quickly pour molten mixture onto an LB agar plate and tilt the plate to evenly distribute the agar. Let sit undisturbed until the agar has gelled.

Once plate has gelled (∼5 min), invert and incubate overnight at 37 °C.

Count phage plaques and determine phage titer in plaque-forming units (PFU·ml−1) using the following formula:

PFU per ml = plaques per plate × volume plated in ml × dilution factor

e.g., if there are 20 plaques when you plated out 1 ml from the 106 dilution, the titer of the phage stock is 2.0 × 107 PFU·ml−1.

2A| Phage isolation and propagation via plate lysate

From a plate lysate plate pick a single phage plaque using a sterile Pasteur pipet.

Resuspend the plaque into a microcentrifuge tube containing 1 ml of filter-sterilized phage diluent (SM buffer) and vortex for 5 min.

Centrifuge at 4,000 × g for 5 min to remove any remaining debris.

Perform plate lysate as described above using the resuspended phage.

After incubation the entire plate should be lysed. Pour ∼5 ml of SM buffer on top of plate and shake gently for 15 min at room temperature.

Collect buffer from top of plate and centrifuge at 4,000 × g for 5 min.

Collect phage lysate and store at 4 °C until clean up.

Optional: Titer the lysate via plaque assay to ensure initial high titer.

2B| Phage propagation via liquid lysate

Grow E. coli bacterial host in LB broth overnight at 37 °C.

Prepare and autoclave 100 ml of LB broth supplemented 0.001 M CaCl2 and MgCl2 added in a 250 ml PYREX® screw cap storage bottle and save for Step 3.

Spike LB broth supplemented with CaCl2 and MgCl2 with 0.1 volumes of overnight bacterial host.

Note: For a phage with a high burst size, such as T3, you may need to double volume of host added.

Incubate with agitation for 1 h at 37 °C.

Add 100 μl of high titer phage lysate (> 108 PFU·ml−1).

Incubate at 37 °C with agitation for ∼5 h or until lysate clears.

Collect phage lysate and store at 4 °C until clean up.

Optional: Titer the lysate via plaque assay to ensure initial high titer.

3| Phage cleanup (0.22 μm filtering and chloroform)

Aliquot phage lysate into 50 ml sterile falcon centrifuge tubes and centrifuge at 4,000 × g for 20 min.

Carefully collect supernatant using a serological pipette and transfer into properly labeled sterile falcon tube.

Filter-sterilize the phage supernatant using a 0.22 μm filter to yield a bacterial cell-free phage lysate.

Add 0.1 volumes of chloroform to the supernatant, vortex, and incubate at room temp for 10 min.

Note: Lipid enveloped phages are sensitive to chloroform and titer can be significantly reduced. If a drop in titer is observed skip this step.

Centrifuge at 4,000 × g for 5 min and transfer supernatant into 250 ml PYREX® screw cap storage bottle and store at 4 °C until concentration.

Optional: Titer the lysate via plaque assay to ensure initial high titer.

4| Phage concentration and wash via ultrafiltration

Add ∼15 ml of phage lysate into the upper reservoir of Amicon filter device.

Centrifuge Amicon at 4,000 × g for ∼5 min.

Note: Centrifugation times will vary based on phage type and titer. It is important not to spin the device dry. If unsure about centrifugation times select a shorter spin time, check the lysate level, and adjust spin times appropriately.

Carefully discard the filtrate into a waste bucket and add another volume of phage lysate to the sample filter cup and repeat centrifugation.

Note: The same device can be used to concentrate large volumes of phage lysate (> 100 ml) achieving an approximate 90% decrease with minimal loss in phage titer.

Repeat step until all phage lysate has been concentrate to < 10 ml.

Add ∼15 ml of SM buffer into the upper reservoir containing concentrate phage lysate and centrifuge at 4,000 × g for ∼5 min to wash phage lysate.

Note: SM buffer was chosen as it is suitable for the long-term storage of T-phage, but any appropriate storage buffer can be used. It is important not to spin the device dry, adjust centrifugation times accordingly.

Repeat wash step and concentrate washed phage lysate to < 10 ml.

Using a pipette, carefully collect phage lysate from the upper reservoir and gently wash the surface of the upper reservoir.

Note: Alternatively the entire device can be vortexed to assist with phage detachment from filter.

Collect < 10 ml of concentrated and purified phage lysate. Titer phage concentrate and record PFU·ml−1.

5| Endotoxin removal (Morrison & Leive, 1975; Szermer-Olearnik & Boratyński, 2015)

Notes: This method is adapted from Szermer-Olearnik & Boratyński (2015), which demonstrates the efficient removal of endotoxins from bacteriophage lysates using water immiscible solvents that are subsequently removed via dialysis. For detailed explanation of the methodology please see Morrison & Leive (1975) and Szermer-Olearnik & Boratyński (2015).

Our adapted method uses a speed vacuum to remove residual organic solvent from phage lysates, instead of the lengthy dialysis washes with similar efficiency.

This step is optional. If you do not require removal of bacterial endotoxins from your phage preparations please go to step 7.

Add 0.4 volumes of 1-octanol to phage concentrate and shake at room temp for 1 h.

Incubate phage concentrate at 4 °C for 1.5 h.

Note: Alternatively concentrate can be chilled in a ice bath for ∼15 min.

Centrifuge at 4,000 × g for 10 min.

Using a syringe, pierce the bottom of the tube and collect the aqueous phase (bottom layer) that contains your phage, and transfer to a sterile 50 ml falcon tube. Do not collect the organic phage (top layer) or interface as this contains endotoxins.

Note: It is best to leave a small residual amount of phage concentrate behind to reduce the transfer of contaminating endotoxins.

Note: If endotoxin removal is not sufficient the method can be repeated to further reduce endotoxins.

6A| Dialysis removal of organic solvent (Szermer-Olearnik & Boratyński, 2015)

Notes: This method is adapted from Szermer-Olearnik & Boratyński (2015) and describes the removal of residual organic solvents from phage lysates by dialysis.

Residual organic solvents disable downstream Pierce™ LAL Chromogenic Endotoxin Quantitation assays and must be removed in order to accurately quantify endotoxin concentrations.

Due to the ionic concentration of phage SM buffer used you may end up with greater than the starting volume.

Pre-wet Spectra-Por® dialysis membrane with sterilized dH2O according to manufacturers instructions, being careful not to contaminate the inside of the tubing.

Load maximum of 10 ml of phage concentrate inside of dialysis tubing and seal tightly.

Dialyze phage concentrate against 2 liters of 25% (v/v) ethanol at 4 °C on a stirring plate for 24 h to remove residual 1-butanol, replacing the 25% ethanol solution four times at 15, 18, 21, and 24 h.

Dialyze phage concentrate against 2 liters of 0.15 M NaCl solution at 4 °C on a stirring plate 24 h to remove residual ethanol, replacing the 0.15 M NaCl solution three times after 15, 19, and 24 h.

Carefully collect phage concentrate by washing the inside of the dialysis tubing. Store concentrate at store 4 °C until phage bank preparation.

Optional: Titer the lysate via plate lysates to ensure high titer. Perform the Pierce™ LAL Chromogenic Endotoxin Quantitation according to manufacturers instructions to obtain quantitative endotoxin levels.

6B| Speed vacuum removal of organic solvent

Notes: This method is a faster alternative to the dialysis method for the removal of residual organic solvents from phage concentrates.

Aliquot ∼1 ml of the phage concentrate equally into microcentrifuge tubes.

Place tubes into a speed vacuum 4 °C, open lids, and centrifuge at 4,000 × g for 3 h.

After speed vacuum, phage concentrate volume should be reduced by approximately 30% and residual 1-octanol evaporated.

Collect phage concentrate and store 4 °C until phage bank preparation.

Optional: Titer the lysate via plaque assay to ensure high titer. Perform the Pierce™ LAL Chromogenic Endotoxin Quantitation according to manufacturers instructions to obtain quantitative endotoxin levels from phage concentrates.

7| Phage bank storage

Dilute phage lysate in SM buffer to generate a high-titer working stock.

Note: This step largely depends on desired concentration and volume of the phage bank. If you require higher titer phage stocks then dilute less or omit dilution, if greater volume is desired then dilute more.

Titer diluted phage concentrate and record PFU·ml−1.

Aliquot phage working stocks into labeled cryo tubes and store at 4 °C.

You now have a bank of homogenous, high titer (up to 1010–11 PFU·ml−1) phage bank for laboratory testing.

From our experience the use of homogenous and standardized phage banks can significantly reduce the experimental variability for the molecular characterization of phages.

Results

Phage purification and concentration

Here we used T4 phage as a representative phage for our PoT phage purification protocol (Fig. 1). We tested the speed and efficiency of the PoT purification protocol (Fig. 2A) compared against the traditional method (Fig. 2B), which involved phage propagation, centrifugation, filtering and purification via CsCl ultracentrifugation and dialysis (Figs. S1 and S2) (Adams, 1959). Both methods started with 110 ml of total lysate and comparable titers. The PoT protocol consisted of phage clean up, concentration and washing via ultrafiltration and resulted in a 93% reduction in volume with a phage recovery efficiency of 85%, while taking just two days to complete. Comparatively, the traditional method, which consisted of phage clean up, CsCl ultracentrifugation and dialysis, resulted in an 80% reduction in volume, with a 4% phage recovery efficiency and taking five days to complete.

Figure 1 Flow chart of the PoT protocol for the production of high titer, homogenous, endotoxin reduced phage banks within two days.

Phage titers (PFU·ml−1) and volumes (mL) were recorded after each step. Endotoxin removal is adapted from Szermer-Olearnik & Boratyński (2015).

Figure 2 A comparison between the PoT protocol and traditional method for the purification and concentration of laboratory phage stocks.

(A) The PoT protocol generated a 10 mL phage stock (7.7 × 1010 PFU·ml−1) with a phage recovery efficiency of 85% within two days. (B) The traditional method generated a 22 mL phage lysate (3.7 × 109 PFU·ml−1) with a phage recovery efficiency of 4% in five days.

Endotoxin content

Phage preparations are often contaminated with bacterial endotoxins and their removal from phage preparations is required for some applications (Merril, Scholl & Adhya, 2003; Chan, Abedon & Loc-Carrillo, 2013). Endotoxin concentrations of T4 lysates from the PoT protocol and traditional method were measured using the LAL chromogenic endotoxin quantitation kits to compare the efficiency of endotoxin removal (Fig. 3). The endotoxin content of the raw lysates were 4 × 104 EU·ml−1, but post clean up neither the PoT nor the traditional method achieved sufficient reductions in endotoxin levels (< 20 EU·ml−1), although the PoT protocol did perform marginally better, with a reduction to 8 × 103 compared to 3 × 104 EU·ml−1 for the traditional method.

Figure 3 Endotoxin concentration (EU·ml−1) of phage preparations as measured by the Pierce™ LAL Chromogenic Endotoxin Quantitation Kit.

Raw lysates had starting endotoxin concentration of 4.7 × 104 EU·ml−1. The PoT protocol and traditional method reduced endotoxin concentrations to 8.3 × 103 and 3.2 × 104 EU·ml−1, respectively. Filter-sterilized SM buffer (0.2 EU·ml−1) is shown as a negative control, dotted line represent the desired 20 EU·ml−1 cut off. Endotoxin concentration data shown are averages of triplicates from a single experiment.

Endotoxin removal

In order to generate phage banks with endotoxin concentrations of < 20 EU·ml−1, we adapted a recently published protocol detailing the removal of endotoxins through extraction with organic solvents and subsequent dialysis washes (Morrison & Leive, 1975; Szermer-Olearnik & Boratyński, 2015). The T4 phage concentrate from our PoT protocol was processed through the endotoxin removal protocol using the organic solvent 1-Octanol (Fig. 4A). After endotoxin removal there was a slight reduction in phage volume and titer, likely due to loss of phage concentrate that was left near the organic layer. Unfortunately, endotoxin quantification at this step was not possible due to residual 1-Octanol in the phage concentrate disabling LAL endotoxin test (Szermer-Olearnik & Boratyński, 2015). In order to remove residual 1-Octanol, phage concentrates were split evenly, processed through dialysis and speed vacuum (5 ml each) and endotoxin levels quantified (Fig. 4B). Following dialysis washes, we were capable of producing a T4 phage concentrate with 14 EU·ml−1, with a 30% increase in volume (likely due to water movement to the higher ionic strength SM buffer), and a phage recovery efficiency of 60% within four days. The major limitation with this method is the lengthy dialysis step required to remove residual 1-Octanol solvents from the phage concentrate, taking upwards of 48 h to complete. We tested a faster speed vacuum method to remove this residual solvent, taking just 3 h to complete and producing a lysate with a slightly increased endotoxin content of 27 EU·ml−1, with a 20% reduction in volume and a phage recovery efficiency of 47%.

Figure 4 PoT protocol processed for the removal of bacterial endotoxins by 1-Octanol and solvent removal by dialysis and speed vacuum.

(A) The PoT protocol processed for endotoxin removal by the dialysis and speed vacuum methods generated a 7.2 mL (3.8 × 1010 PFU·ml−1) and 4.2 mL (4.7 × 1010 PFU·ml−1) phage banks, respectively. (B) Endotoxin concentrations of dialysis (14 EU·ml−1) and speed vacuum (27 EU·ml−1) processed PoT phage banks. Quantification of 1-Octanol treated concentrate was not possible due to residual solvent disabling the LAL quantification test, dotted line represent the desired 20 EU·ml−1 cut off.

Phage banks and storage

Phage banks were produced from purified concentrates to provide homogenous, high titer, endotoxin reduced phage stock for repeat molecular testing and characterization. As such it was important to ensure the stability of phage banks under repeated laboratory use. We tested a range of conditions for the shot-term storage and repeated use of T4 phage banks, including 4 °C in SM buffer, liquid nitrogen in 50% v/v glycerol, liquid nitrogen in 5% v/v DMSO, −20 °C in 50% v/v glycerol and −80 °C in 50% v/v glycerol (Fig. 5A). Phage bank storage at 4 °C in SM buffer showed minimal reduction in titer for short-term storage of high-use phage banks. Freezing of high-use phage banks is not recommended, as the repeated freeze-thaw cycles from the frozen preparations likely damaged phage, reducing titer. If long-term storage of phage banks is required, we recommend storage in liquid nitrogen with 5% v/v DMSO with minimal freeze-thaws, as this showed lowest loss of phage titer over a 3-month period. However, prior to use of frozen phage banks, it is advised to wash phage lysate with SM buffer through an ultrafiltration unit to remove any residual DMSO. Finally, T4 phage morphology and structure following PoT protocol was determined by transmission electron microscopy (TEM) (Fig. 5B). T4 phage showed tails that were not contracted and intact capsid structures, indicating that PoT protocol did not negatively impact phage structure and viability.

Figure 5 Stability of T4 phage processed through PoT under high-use laboratory conditions.

(A) The mean titer of T4 phage stored at 4 °C in SM buffer, liquid nitrogen (LN2) in 5% (v/v) DMSO, LN2 in 50% (v/v) glycerol, −20 °C in 50% glycerol and −80 °C in 50% glycerol. Phage stocks were tittered in duplicate under high-use conditions (full titers Fig. S4). (B) Negative stain transmission electron micrograph of T4 phage processed through the PoT protocol (scale bar = 100 nm).

Applicability of Phage on Tap protocol

As the PoT protocol is based on an ultrafiltration methodology, which we believe is broadly applicable to other tailed phages that can be isolate and cultured to high titer. To test this, we processed four tailed phages, including T4Δhoc–a Myoviridae infecting E. coli; T3–a Podoviridae infecting E. coli; T5–a Siphoviridae infecting E. coli; and Spp1–a Siphoviridae infecting the Gram-positive Bacillus subtilis, through the PoT protocol (Fig. 6). Phages T3, T5 and Spp1 were propagated by plate lysates, while T4Δhoc phage was propagated by liquid lysate. All phages were processed through to a concentrate with a > 50% phage recovery efficiency, followed by 1-Octanol endotoxin removal and processing by dialysis and speed vacuum. All phages showed a large reduction in endotoxin concentrations, but none were below the desired < 20 EU·ml−1 with the exception of the Spp1 phage, which infects the Gram-positive Bacillus subtilis bacterial host. The Spp1 phage lysate produced an already low endotoxin lysate that we further reduced to < 1 EU·ml−1. The dialysis method was more efficient than speed vacuum at removing bacterial endotoxins, but took 48 h to complete. Comparatively, the speed vacuum method took only 3 h to complete, producing a phage concentrate with an average 2-fold higher endotoxin concentration than the dialysis method with a comparable titer. Overall, all of the phages processed through the PoT protocol generated high titer, homogenous phage banks with significantly reduced endotoxin levels.

Figure 6 Applicability of PoT for the generation of high-titer, homogenous, endotoxin reduced phage banks from diverse phages.

(A) T4Δhoc phage that infects E. coli bacterial host propagated by liquid lysate (2.2 × 1010 PFU·ml−1). (B) T3 phage that infects E. coli bacterial host propagated by plate lysate (1.1 × 1010 PFU·ml−1). (C) T5 phage that infects E. coli bacterial host propagated by plate lysate (2 × 109 PFU·ml−1). (D) Spp1 phage that infects B. subtilis Gram-positive bacterial host propagated by plate lysate (1.5 × 109 PFU·ml−1). (E) Endotoxin concentrations of raw lysates, PoT concentrate, dialysis and speed vacuum treated phage banks, dotted line represent the desired 20 EU·ml−1 cut off. Overall, a decrease in volume (ml) of the lysates was observed after each step of the procedure.

Discussion

Traditional methods for the isolation and generation of phages lysates often involve centrifugation, precipitation with PEG, ultracentrifugation in CsCl gradients, followed by dialysis and storage. These procedures are lengthy and time consuming and generate phage lysates with variable titer, endotoxin, and ionic concentration. Here we present a fast and efficient method to produce homogeneous phage banks for laboratory testing and molecular characterization. Our method focused on T4 phage, but is broadly applicable to other phages that can be isolated in high titer (> 109 PFU·ml−1). Specifically, the T4 phage was propagated with E. coli bacterial host in liquid lysate, purified by centrifugation, 0.22 μm filtration and chloroform treatment, concentrated by ultrafiltration centrifugation, and stored in large phage banks at 4 °C in buffer. Unfortunately, neither the PoT protocol nor the traditional methods were effective at significantly reducing bacterial endotoxins from phage preparations, and additional purification steps were required.

Numerous methods and commercially available kits are available for the removal of bacterial endotoxins (Merril et al., 1996; Boratyński et al., 2004; Merabishvili et al., 2009; Oślizło et al., 2011; Branston, Wright & Keshavarz-Moore, 2015), yet many of these are either highly specific, time consuming, laborious, or expensive. Szermer-Olearnik & Boratyński (2015) recently proposed the use of an organic solvent to successfully reduce bacterial endotoxin from phage lysates (< 20 EU·ml−1). The method is cheap, broadly applicable, and capable of endotoxin removal regardless of initial variations in titer and endotoxin levels, but does rely on multiple dialysis steps that are time consuming (Morrison & Leive, 1975; Szermer-Olearnik & Boratyński, 2015). It may be possible to shorten the dialysis method through more frequent changes in buffer solution, although this needs to be confirmed. Here we modify this method by replacing the long dialysis steps with a speed vacuum step for the removal of residual 1-octanol from the lysates. The speed vacuum modification was not as efficient as the dialysis step for the removal of bacterial endotoxins, but was significantly shorter to complete–taking just 3 h compared to the 48 h required for dialysis. Using this modification we are able to produce homogenous, high-titer, endotoxin-reduced phage banks within two days.

Phage are generally quite stable at high concentrations when stored at 4 °C, free of bacterial debris, protected from light exposure, and in appropriately buffered solution with a pH between 5–9 (Adams, 1959; Clark, 1962; Wommack et al., 1996; Jończyk et al., 2011). Although most tailed and filamentous phages can be easily stored under these conditions for 5–10+ years, it is always best to determine the optimal storage conditions for each phage of interest. Phage are generally sensitive to freezing, thawing, and lyophilization and titer is known to vary with storage time (Clark, 1962; Clark & Klein, 1966; Clark & Geary, 1973; Jończyk et al., 2011). After monitoring the degradation of numerous phages for over 21 years, it was shown that phage storage at 4 °C and −80 °C was suboptimal compared to storage in liquid nitrogen (−196 °C) (Ackermann, Tremblay & Moineau, 2004). However, due to their diversity and the difficulties associated with accurate monitoring, the optimal conditions for long-term storage of phage (> 10 years) remains uncertain. The purpose of PoT protocol was to generate homogenous, high-titer phage banks for regular and consistent laboratory use. For this purpose the storage of phage in SM buffer at 4 °C over three months showed no noticeable depreciation in titer, which is consistent with previously described short-term storage conditions.

The PoT protocol described here takes two days to produce high titer, homogenous, endotoxin reduced phage banks for molecular characterizations. In comparison, the traditional method can take upwards of five days to complete, is laborious, and generates a lower titer phage lysate with high endotoxin levels. Traditional phage propagations suffer from further variability in titer, salts, and bacterial contaminants between successive lysates. Eliminating this variability through the use of PoT phage banks has been critical for the accurate analysis and molecular investigations of phage within the context of eukaryotic hosts (Barr et al., 2013; Barr et al., 2015). Phage banks can be easily stored at 4° C in SM buffer while maintaining their viability under continuous high-use. The PoT protocol is efficient, can be easily completed in the laboratory, may be among the least costly, and generates large homogenous phage banks for repeated use.

Supplemental Information

Supplemental Information 1 Phage on Tap Supplemental Material.

Materials and Methods, Figs. S1 and S2.

Click here for additional data file.

We thank the San Diego State University TEM Facility for the help with the TEM analyses.

Additional Information and Declarations

Competing Interests

Author Contributions

Data Deposition

The authors declare that they have no competing interests.

Natasha Bonilla conceived and designed the experiments, performed the experiments, analyzed the data, contributed reagents/materials/analysis tools, wrote the paper, prepared figures and/or tables, reviewed drafts of the paper.

Maria Isabel Rojas performed the experiments, reviewed drafts of the paper.

Giuliano Netto Flores Cruz performed the experiments, reviewed drafts of the paper.

Shr-Hau Hung contributed reagents/materials/analysis tools, reviewed drafts of the paper.

Forest Rohwer conceived and designed the experiments, contributed reagents/materials/analysis tools, reviewed drafts of the paper.

Jeremy J. Barr conceived and designed the experiments, performed the experiments, analyzed the data, contributed reagents/materials/analysis tools, wrote the paper, prepared figures and/or tables, reviewed drafts of the paper.

The following information was supplied regarding data availability:

The raw data has been supplied as Supplemental Dataset Files.

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
