# Peer review of "Phage on tap–a quick and efficient protocol for the preparation of bacteriophage laboratory stocks"

_PeerJ, doi:10.7717/peerj.2261_

## Round 0.1 · original submission · Minor Revisions

In addition to the minor comments from the two reviewers, I have the following specific notes and edits:

Line 17 and others: is there supposed to be a dot between PFU and ml or EU and ml?
Line 40: please remove the ; between involve and centrifugation
Line 44: please change to “and affect phage recovery and/or viability.”
Line 58: please be consistent with your use of / or -1 to denote per.
Fig. 2: Please put A and B on the same scale for PFU ml-1 so they are more easily comparable
Fig. 3: please indicate how many times these assays were performed so as to give a measure of variation.

Reviewer 1 ·

Basic reporting

no comment

Experimental design

The authors suggest that dialysis is a superior mechanism for the removal of the organic solvent, but delays the overall procedure. Can you speed the dialysis time with more frequent changes? I was under the impression that the bulk of the exchange in dialysis between 10ml to 2L (or, even better to 4L) occurred in the first hour—(see the thermo-fisher literature on dialysis). I think more rapid changes early on would lead to faster dialysis overall, and you might finish in as few as four hours for each for the ethanol and salt parts. You may also be losing titer due to phage particles sticking to the membrane over that long of a time period (perhaps not with T4, but others certainly will). I would like to see the data on dialysis times and residual chemicals if the authors have already empirically determined that this long of a time frame is necessary.

https://www.thermofisher.com/us/en/home/life-science/protein-biology/protein-biology-learning-center/protein-biology-resource-library/pierce-protein-methods/dialysis-methods-protein-research.html

Validity of the findings

No comment

Additional comments

I find this paper within the scope of the journal, methodologically sound, and clearly presented. And intriguing!

I have only a few comments:


There are numerous references to “top agar” as a procedure rather than as a material. I believe this is not correct according to the literature, as top (or soft) agar can be used for a number of different reasons in phage experiments. If the goal of the procedure is to generate a high titer lysate, the correct name for the procedure is “plate lysate” (as it then parallels “liquid lysate”), if it is to generate countable plaques for titering, the correct name is “plaque assay”.


I would like some clarity about the use of the speed vac:
Line 278: what temp do you run the speed vac at? Do you have temperature control? Do you make it warmer to evaporate the solvent faster? Do you lose titer if you do?

·

Basic reporting

No Comments

Experimental design

No Comments.

Validity of the findings

No Comments.

Additional comments

Summary:
Bonilla et al. report a combinatorial protocol for Phage on Tap, a method for rapidly obtaining high titer phage lysates that have been amplified and purified of some of the troubling bacterial backwash that comes along when lysates are made. The protocol is detailed and handy and this is an important contribution in an era when interest in bacteriophages is increasing.

Specific Notes and edits:

Line 143: is recommended to use appropriated growth and propagation conditions
I enjoy the idea of appropriated growth and you should not edit this.

In Figure 2 the left side Y-axes are different between the PoT and traditional methods while the right side Y-axes are the same. Wouldn’t this be more effective if this were the same?

If Fig 6. Would love to see some sort of normality in these axes. This is all over the show. I sort of get the reason: you are more interested in the relative decrease as the lysate goes through each procedure than you are in the ABSOLUTE change, which is just a result of whatever starting material you have. Perhaps you could explain this however in the figure legend so that readers aren’t scratching their heads about what is going on. Alternatively you could just graph the proportion of volume lost and the relative titer that way the plots could be standardized.

---

## Round 0.2 · accepted · Accept

Thanks for making those changes.